# Sugar is an endogenous cue for juvenile-to-adult phase transition in plants

Sha Yu[1,2], Li Cao[2,3], Chuan-Miao Zhou[1], Tian-Qi Zhang[1,2], Heng Lian[1], Yue Sun[4], Jianqiang Wu[5], Jirong Huang[1], Guodong Wang[3], Jia-Wei Wang[1]*

[1]National Key Laboratory of Plant Molecular Genetics, Institute of Plant Physiology and Ecology, Shanghai Institutes for Biological Sciences, Shanghai, China; [2]Graduate School of Chinese Academy of Sciences, Beijing, China; [3]State Key Laboratory of Plant Genomics and National Center for Plant Gene Research, Institute of Genetics and Developmental Biology, Chinese Academy of Sciences, Beijing, China; [4]School of Life Sciences, East China Normal University, Shanghai, China; [5]Key Laboratory of Economic Plants and Biotechnology, Kunming Institute of Botany, Kunming, China

**Abstract** The transition from the juvenile to adult phase in plants is controlled by diverse exogenous and endogenous cues such as age, day length, light, nutrients, and temperature. Previous studies have shown that the gradual decline in microRNA156 (miR156) with age promotes the expression of adult traits. However, how age temporally regulates the abundance of miR156 is poorly understood. We show here that the expression of miR156 responds to sugar. Sugar represses miR156 expression at both the transcriptional level and post-transcriptional level through the degradation of miR156 primary transcripts. Defoliation and photosynthetic mutant assays further demonstrate that sugar from the pre-existing leaves acts as a mobile signal to repress miR156, and subsequently triggers the juvenile-to-adult phase transition in young leaf primordia. We propose that the gradual increase in sugar after seed germination serves as an endogenous cue for developmental timing in plants.

*For correspondence: jwwang@sibs.ac.cn

**Competing interests:** The authors declare that no competing interests exist.

**Reviewing editor**: Richard Amasino, University of Wisconsin, United States

## Introduction

After seed germination, plants undergo two developmental transitions: juvenile-to-adult and adult-to-reproductive (*Bäurle and Dean, 2006*). The transition from the juvenile to adult phase is marked by acquisition of reproductive competence and changes in leaf morphology (*Poethig, 2010*). The adult to reproductive transition, also known as flowering, transforms the identity of the shoot apical meristem from vegetative into inflorescence. Physiological and genetic studies have demonstrated that both developmental transitions are regulated not only by environmental signals such as day length, light intensity, and ambient temperature, but also by endogenous signals transmitted by plant hormones and age.

microRNA156 (miR156), which targets SQUAMOSA PROMOTER BINDING PROTEIN-LIKE (SPL) transcriptional factors, provides an endogenous age cue for developmental timing in plants (*Poethig, 2010*). The expression of miR156 decreases over time, with a concomitant rise in SPL level (*Wu and Poethig, 2006*; *Wang et al., 2009*). Overexpression of miR156 prolongs the juvenile phase, whereas a reduction in miR156 level results in an accelerated expression of adult traits (*Wu and Poethig, 2006*; *Wu et al., 2009*). SPL promotes the juvenile-to-adult phase transition and flowering through activation of miR172 and MADS-box genes (*Wang et al., 2009*; *Wu et al., 2009*; *Yamaguchi et al., 2009*; *Jung et al., 2011*). Very recently, defoliation experiments and expression analyses demonstrated that the repression of miR156 in the leaf primordia is mediated by a mobile signal(s) derived from the pre-existing leaves (*Yang et al. 2011*). However, the identity of this signal is still unknown.

**eLife digest** Like animals, plants go through several stages of development before they reach maturity, and it has long been thought that some of the transitions between these stages are triggered by changes in the nutritional status of the plant. Now, based on experiments with the plant *Arabidopsis thaliana*, Yu et al. and, independently, Yang et al. have provided fresh insights into the role of sugar in 'vegetative phase change'—the transition from the juvenile form of a plant to the adult plant.

The new work takes advantage of the fact that vegetative phase change is controlled by two genes that encode microRNAs (MIRNAs). *Arabidopsis* has eight *MIR156* genes and both groups confirmed that supplying plants with sugar reduces the expression of two of these—*MIR156A* and *MIR156C*—while sugar deprivation increases their expression. Removing leaves also leads to upregulation of both genes, and delays the juvenile-to-adult transition. Given that this effect can be partially reversed by providing the plant with sugar, it is likely that sugar produced in the leaves—or one of its metabolites—is the signal that triggers the juvenile-to-adult transition through the reduction of miR156 levels.

Yu and co-workers confirmed that sugar also reduces the expression of *MIR156* in tobacco, moss, and tomato plants, suggesting that this mechanism is evolutionarily conserved. Consistent with the work of Yang and colleagues, Yu and co-workers revealed that sugar is able to reduce the transcription of *MIR156A* and *MIR156C* genes into messenger RNA. Moreover, they showed that sugar can also suppress *MIR156* expression by promoting the breakdown of *MIR156A* and *MIR156C* primary messenger RNA transcripts.

The work of Yu et al. and Yang et al. has thus provided key insights into the mechanisms by which a leaf-derived signal controls a key developmental change in plants. Just as fruit flies use their nutritional status to regulate the onset of metamorphosis, and mammals use it to control the onset of puberty, so plants use the level of sugar in their leaves to trigger the transition from juvenile to adult forms.

In addition to being essential as prime carbon and energy sources, sugars also play critical roles as signaling molecules (*Rolland and Sheen, 2005*; *Smeekens et al., 2010*). In *Arabidopsis thaliana*, diverse sugar signals are perceived and transduced through a glucose sensor, HEXOKINASE1 (HXK1). HXK1 exerts its regulatory function through distinct molecular mechanisms including transcriptional activation, translational inhibition, mRNA decay, and protein degradation (*Rolland and Sheen, 2005*). Analyses of two catalytic inactive *HXK1* alleles further indicate that the signaling activity of HXK1 is uncoupled from its catalytic activity (*Moore et al., 2003*). Recently, a nuclear HXK1 complex has been identified (*Cho et al., 2006*). In this complex, HXK1 binds to two unconventional partners, the vacuolar $H^+$-ATPase B1 (VHA-B1) and the 19S regulatory particle of a proteasome subunit (RPT5B). Since neither VHA-B1 nor RPT5B has DNA binding capacity, the precise molecular mechanism by which this nuclear-localized HXK1 complex regulates gene expression remains unanswered. In addition to the HXK1-dependent pathway, some glucose-responsive genes are regulated through an HXK1-independent pathway. For instance, the expression of the genes encoding chalcone synthase, phenylalanine ammonia-lyase, and asparagine synthase responds to glucose signaling in the absence of HXK1 (*Xiao et al., 2000*).

Here, we performed expression and mutant analyses to identify the upstream regulator of miR156. Our results demonstrate that the expression of miR156 quickly responds to sugar. Sugar reduces miR156 abundance through both transcriptional repression and transcript degradation. Thus, gradual accumulation of sugar after seed germination leads to a reduced level of miR156, which promotes the juvenile-to-adult phase transition in plants.

## Results

### *MIR156A* and *MIR156C* have dominant roles within the *MIR156* gene family

The transition from juvenile to adult phase in Arabidopsis is accompanied by changes in vegetative morphology. Under long day conditions, the wild type Arabidopsis plants switch from the juvenile to the adult phase from the fifth or sixth leaf. The juvenile leaves are round, smooth on their margins, and

barely develop trichomes (leaf hairs) on the abaxial side (lower side). By contrast, the adult leaves are elongated, serrated, and produce abaxial trichomes (*Wu et al., 2009*).

In the Arabidopsis genome, miR156 is encoded by eight coding loci (*MIR156A–MIR156H*) (*Reinhart et al., 2002*). To understand which locus or loci play important roles within this gene family, we identified all available *MIR156* transfer-DNA (T-DNA) knockout plants (*Samson et al., 2002*; *Alonso et al., 2003*; *Woody et al., 2007*; *Figure 1A* and *supplementary file 1A*). Due to functional redundancy, none of these mutants exhibited visible developmental defects (data not shown). One of the double mutants, *mir156a mir156c*, displayed a similar, but weak phenotype as the transgenic plant expressing a target mimicry from the constitutively active *35S* promoter (*35S::MIM156*), which reduced miR156 activity (*Figure 1D*; *Franco-Zorrilla et al., 2007*; *Todesco et al., 2010*). RNA gel blot demonstrated that the amount of miR156 was moderately decreased in *mir156a mir156c* in comparison with the wild type (*Figure 1B*). Accordingly, the transcript levels of two miR156-target genes, *SPL3* and *SPL9*, were much higher in *mir156a mir156c* than in the wild type (*Figure 1C*).

Compared to the wild type, the *mir156a mir156c* mutant had a shortened juvenile phase. The appearance of abaxial trichomes in *mir156a mir156c* was accelerated by 2.1 plastochrons (*Figure 1E*). In addition, the length-to-width ratios of the blades in *mir156a mir156c* were much closer to those of the adult leaves in the wild type (*Figure 1F*). Furthermore, *mir156a mir156c* flowered earlier than the wild type (*Figure 1E*). Taken together, these results indicate that *MIR156A* and *MIR156C* have dominant roles within the miR156 family in Arabidopsis.

## Sugar represses *MIR156* expression

To elucidate the molecular mechanism by which the level of miR156 is regulated by age, we performed time course expression assays on miR156 and the primary transcripts of *MIR156A* and *MIR156C* (*pri-MIR156A* and *pri-MIR156C*) by RNA gel blot and quantitative real-time PCR (qRT-PCR). We collected plants grown under long day conditions for 8, 9, and 16 days. As previously reported, the abundance of miR156 gradually declined (*Figure 2A,B*; *Wu and Poethig, 2006*; *Wang et al., 2009*). Interestingly, the transcript levels of *pri-MIR156A* and *pri-MIR156C*, but not mature miR156, exhibited damped oscillations with the highest level in the morning and lowest before dark (*Figure 2C,E*; *Figure 2—figure supplement 1*). To test whether this expression pattern is generated by the circadian clock, we grew wild type plants for 5 days in long day conditions, and then transferred them to a constant light condition. After the transfer, the oscillating expression pattern of *pri-MIR156A* and *pri-MIR156C* was no longer observed (*Figure 2D,F*), demonstrating a negligible effect of the circadian clock on miR156 expression.

In addition to the circadian clock, endogenous carbohydrates are also able to trigger the oscillation of RNA transcripts (*Bläsing et al., 2005*). To test this possibility, we carried out sugar treatment assays. Five-day-old seedlings grown in 1/2 Murashige and Skoog (MS) liquid media were treated with sugars, including two disaccharides (maltose and sucrose) and two hexoses (glucose and fructose). The break-down of maltose results in two glucose molecules, whereas hydrolysis of sucrose produces glucose and fructose. The abundance of *pri-MIR156A* and *pri-MIR156C* was greatly reduced after 1 day of treatment with 50 mM sucrose, glucose, or maltose (*Figure 3A*). A reduction in *pri-MIR156A* or *pri-MIR156C* was not detected when the seedlings were treated with the same concentration of mannitol or 3-O-methyl-glucose (3-OMG), suggesting that the repression of *pri-MIR156* by sugars is not due to osmotic stress. Consistent with the reduction in *pri-MIR156* levels, mature miR156 was decreased after 1 day of sugar treatment (*Figure 3A*; *Figure 3—figure supplement 1*). Accordingly, the transcript levels of miR156-targeted genes, *SPL9* and *SPL15*, were markedly increased (*Figure 3B*).

To monitor how fast miR156 responds to sugar, wild type seedlings were treated with glucose, sucrose, maltose, fructose, or mannitol for 30 min. A reduction of about 40% in *pri-MIR156C* was observed in the seedlings treated with glucose, sucrose, or maltose, while the level of *pri-MIR156C* was not altered in those treated with fructose or mannitol (*Figure 3C*). These results, together with the fact that glucose is the common hydrolytic product shared by sucrose and maltose, suggest that glucose plays a major role in repressing miR156.

To determine whether all the miR156 coding genes are repressed by sugar, we analyzed the expression of their primary transcripts. *pri-MIR156G* and *pri-MIR156H* were not readily amplified, probably

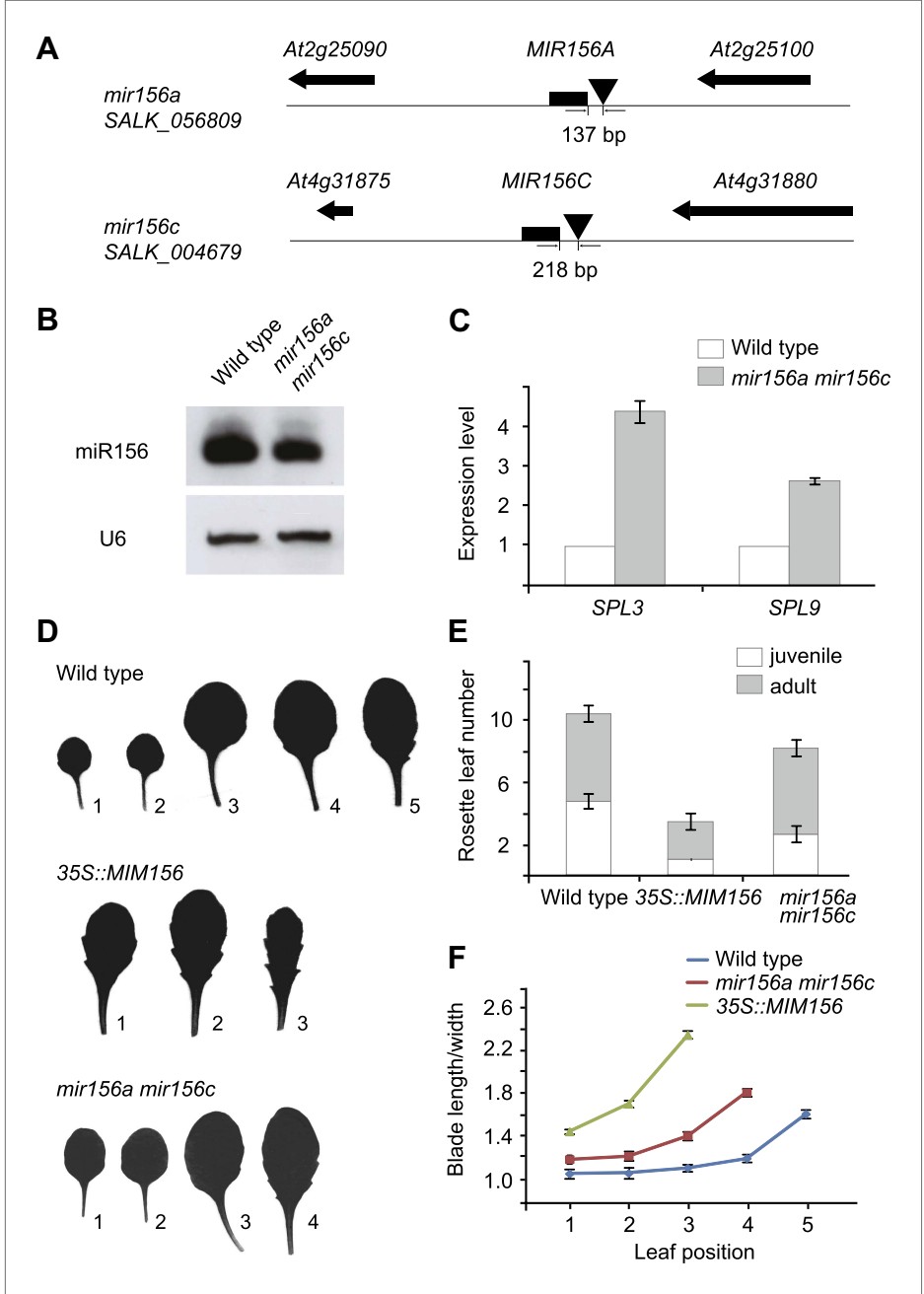

**Figure 1**. Phenotypic analyses of the *mir156a mir156c* double mutant. (**A**) *MIR156A* and *MIR156C* genomic regions. Arrowheads mark T-DNA insertion sites. T-DNAs are inserted 137 bp and 218 bp upstream of the stem-loops of *MIR156A* and *MIR156C*, respectively. (**B**) Expression of miR156 in the wild type and the *mir156a mir156c* double mutant. U6 was monitored as loading control. (**C**) Expression of *SPL3* and *SPL9* in the wild type and the *mir156a mir156c* double mutant. The expression level in the wild type was set to 1.0. (**D**) Leaf morphology of wild type, *mir156a mir156c*, and *35S::MIM156* plants. The leaves were detached and scanned. The numbers indicate leaf positions. (**E**) The number of juvenile and adult leaves. n=12. (**F**) The length-to-width ratio of the blade. Fully expanded leaves were detached and scanned. The length and width of blades were measured. n=12. Error bars indicate SE.

due to their very low expression level (data not shown). The expression of other *pri-MIR156* transcripts except *pri-MIR156B* was reduced after glucose treatment (***Figure 3D***).

To confirm the role of sugar in miR156 expression, we performed a sugar starvation experiment. Five-day-old wild type seedlings were transferred to 1/2 MS liquid media free of sugar and kept in the dark for

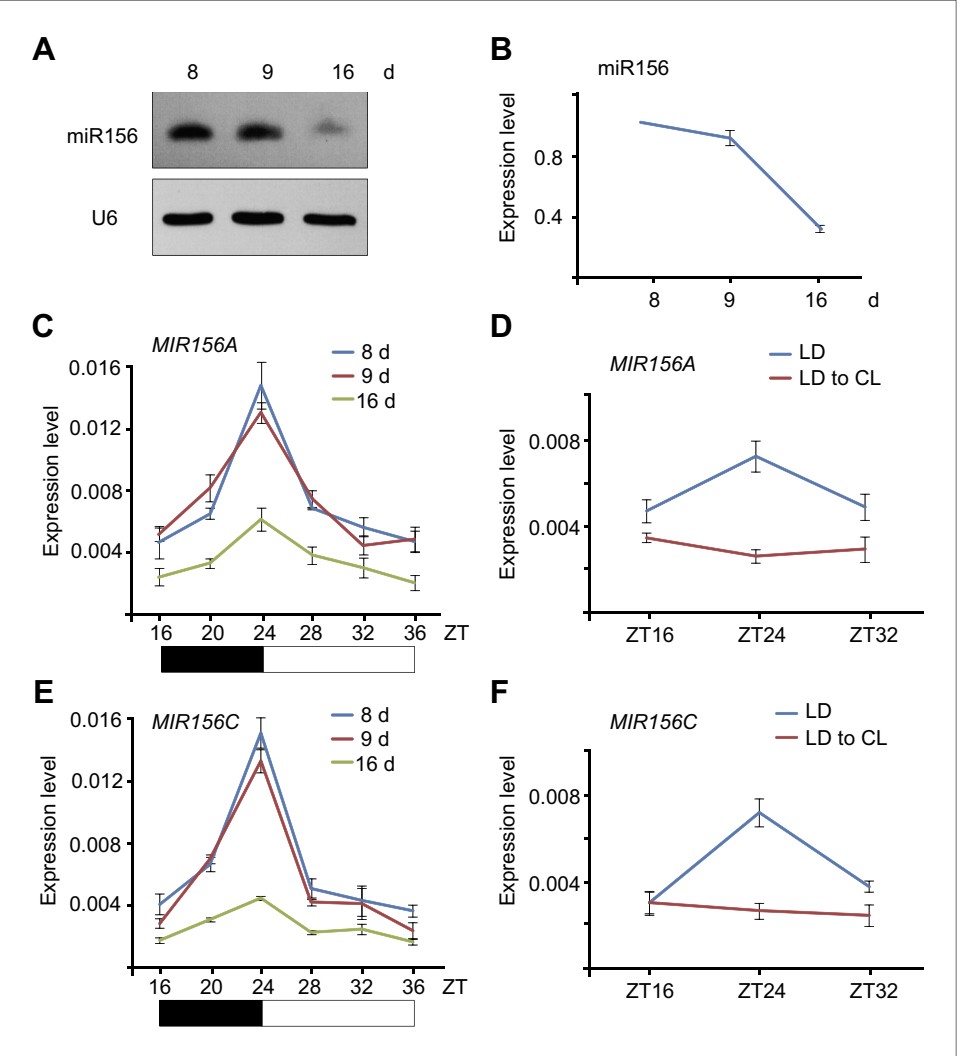

**Figure 2**. Expression of miR156. (**A** and **B**) Accumulation of miR156 in 8-, 9-, and 16-day-old long day plants. Expression of miR156 was analyzed by small RNA blot (**A**) and qRT-PCR (**B**). The plants were collected at Zeitgeber time (ZT) 24. The expression level of miR156 in 8-day-old seedlings was set to 1. (**C** and **E**) Expression of *pri-MIR156A* (**C**) and *pri-MIR156C* (**E**). The plants were collected every 4 hr and subjected to qRT-PCR analyses. Black and white boxes indicate dark and light conditions, respectively. (**D** and **F**) Expression of *pri-MIR156A* (**D**) and *pri-MIR156C* (**F**) during the shift from long day (LD) to constant light (CL) conditions. Five-day-old wild type seedlings were shifted from long day to constant light conditions. The seedlings were collected at ZT 16, 24, and 32.

The following figure supplements are available for figure 2:

**Figure supplement 1**. Expression pattern of miR156.

---

2 days. Compared to the seedlings grown in 1/2 MS liquid media supplemented with sugar under normal light conditions, the sugar-depleted seedlings exhibited a higher expression level of miR156 (*Figure 3E*).

To investigate whether sugar specifically represses miR156, we analyzed the expression of other miRNA primary transcripts, including *pri-MIR159A*, *pri-MIR159B*, and *pri-MIR165A*. The levels of all these transcripts were not reduced after sugar treatment (*Figure 3F*).

## Sugar promotes the juvenile-to-adult phase transition

A recent study has shown that the juvenile-to-adult phase transition is mediated by a leaf-derived mobile signal that represses the expression of miR156 in young leaf primordia (*Yang et al. 2011*).

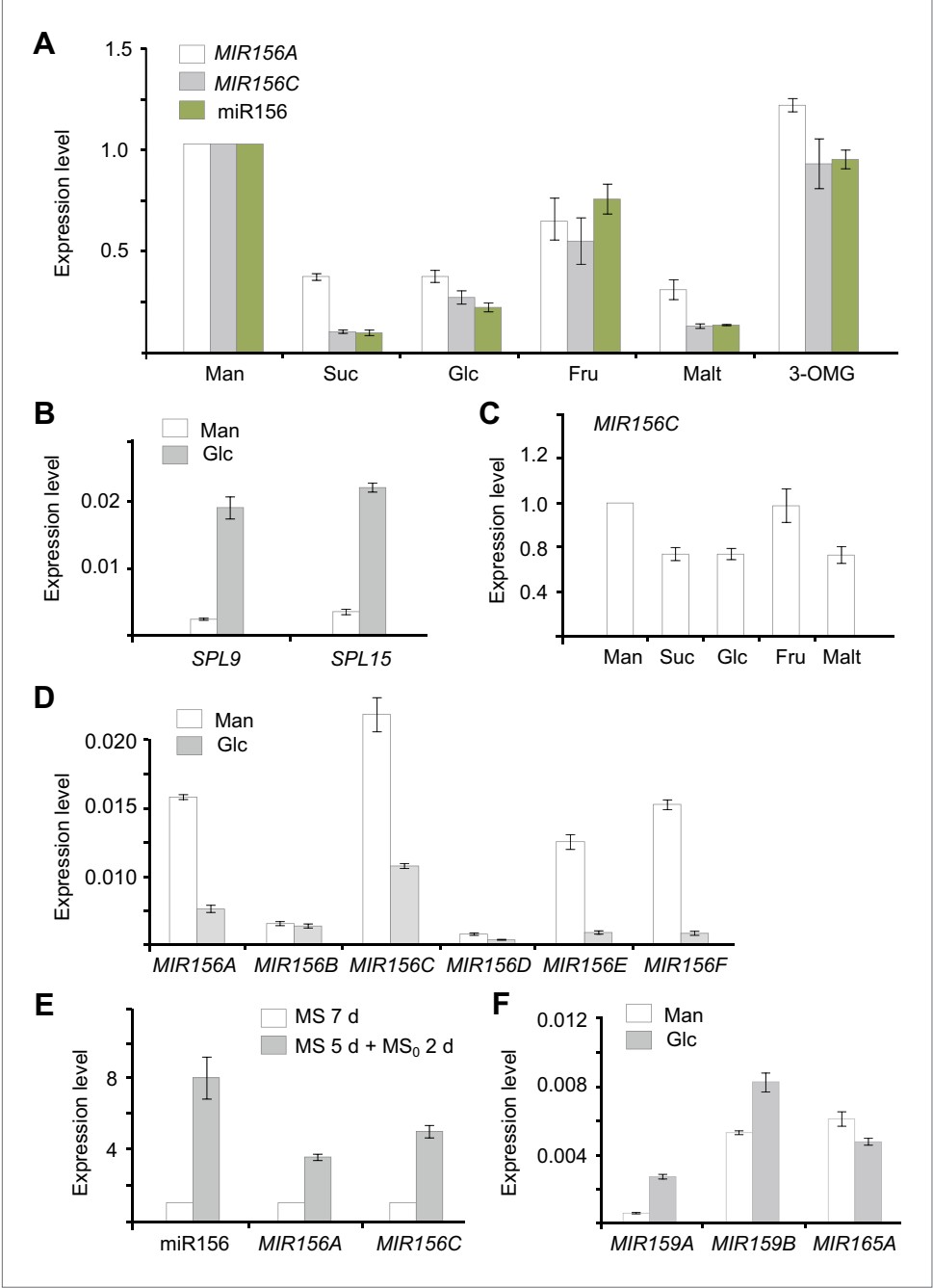

**Figure 3**. Sugar represses miR156. (**A**) Expression of miR156, *pri-MIR156A*, and *pri-MIR156C* in response to sugar. Five-day-old wild type seedlings in 1/2 Murashige and Skoog (MS) liquid media were treated with 50 mM sucrose (Suc), glucose (Glc), fructose (Fru), maltose (Malt), or mannitol (Man) for 1 day. (**B**) Expression of *SPL9* and *SPL15* in response to sugar treatment. Five-day-old wild type seedlings were treated with 50 mM Man or Glc for 1 day. (**C**) *pri-MIR156C* quickly responds to sugar. Five-day-old wild type seedlings were treated with sugar for 30 min. The expression level in the mannitol-treated samples was set to 1. (**D**) Expression of *pri-MIR156* transcripts. Five-day-old wild type seedlings in 1/2 MS liquid media were treated with 50 mM glucose or mannitol for 1 day. (**E**) Expression of miR156 and *pri-MIR156C* during sugar starvation. Five-day-old wild type seedlings in 1/2 MS liquid media supplemented with 50 mM sucrose were transferred to 1/2 MS media without sucrose ($MS_0$). The seedlings were grown for another 2 days and then subjected to expression analyses. Seven-day-old seedlings in 1/2 MS liquid media supplemented with 50 mM sucrose were used as control. (**F**) Expression of other *pri-MIRNA* transcripts. Five-day-old wild type seedlings in 1/2 MS liquid media were treated with 50 mM glucose or mannitol for 1 day.
*Figure 3. Continued on next page*

*Figure 3. Continued*

The expression levels of *pri-MIR156* and miR156 were normalized to those of *TUBULIN* (*TUB*). In the sugar treatment assays, 50 mM sugars were added at Zeitgeber time 12.

The following figure supplements are available for figure 3:

**Figure supplement 1**. Sugar represses miR156.

Given the fact that sucrose is able to move within plants through the vascular tissues (*Truernit 2001*) and that sucrose as well as its hydrolytic product, glucose, repress the expression of miR156, we speculated that sugar is a potential candidate for this mobile signal. To test this hypothesis, we first investigated the relationship between sugar content and the level of miR156 in vivo. Under long day conditions, Arabidopsis plants show a rapid life cycle with very short juvenile and adult phases. For this reason, we grew wild type plants under short day conditions to extend the vegetative phase. Then 15-day-old (in the juvenile phase) and 60-day-old (in the adult phase) plants were collected at Zeitgeber time (ZT) 16. Expression analyses demonstrated that miR156 was highly abundant in 15-day-old plants but less so in 60-day-old plants (*Figure 4A*). In contrast to this expression pattern, 60-day-old plants exhibited a higher level of glucose, fructose, and sucrose than 15-day-old plants (*Figure 4B*). These results are consistent with our findings that sugar represses miR156 and indicate an inverse correlation between the level of miR156 and endogenous sugar content in vivo.

We then performed defoliation assays. The blades of the first two leaves of 7-day-old wild type seedlings were manually removed. Then 50 mM sucrose or mannitol (as control) was applied to the petioles of the defoliated leaves (*Figure 4C*). Consistent with the previous report (*Yang et al. 2011*), the removal of the first two leaves resulted in an increased level of miR156 in the shoot apices (*Figure 4F*). The expression of adult-specific traits was accordingly delayed. Compared to intact plants, the production of abaxial trichomes in the defoliated plants was delayed by 1.0 plastochrons (*Figure 4D*), and the increase in the length-to-width ratio of the lamina was slower (*Figure 4E*).

Sucrose application partially suppressed the delay in the juvenile-to-adult phase transition caused by defoliation. The sucrose-treated plants produced the abaxial trichomes 0.8 plastochrons later than intact wild type plants, but 1.6 plastochrons earlier than the mannitol-treated plants (*Figure 4D*). In addition, the length-to-width ratios of the fifth, seventh, and ninth leaves in the sucrose-treated plants were higher than those in the mannitol-treated plants (*Figure 4E*). In agreement with these phenotypic differences, the expression of miR156 was reduced in the apices of the sucrose-treated plants but not in those treated with mannitol (*Figure 4F*).

## A reduced photosynthetic rate delays the juvenile-to-adult phase transition

To confirm the role of sugar in the juvenile-to-adult phase transition, we analyzed the Arabidopsis *cao/chlorina1* (*ch1*) mutant. A mutation in *CAO/CH1* (At1g44446), which encodes chlorophyll (Chl) *a* oxygenase, causes a reduced level of Chl *b* and low efficiency of photosynthesis (*Espineda et al., 1999*). Compared to the wild type, the *cao/ch1* mutant developed smaller pale green leaves and had a prolonged juvenile phase (*Figure 5—figure supplement 1*). The rosette leaves in the *cao/ch1* mutant were rounder than those in the wild type plant (*Figure 5A,B*). Additionally, the appearance of abaxial trichomes in the *cao/ch1* mutant was delayed (*Figure 5C*). Expression analyses indicated that higher levels of miR156 accumulated in the *cao/ch1* mutant than in the wild type plant (*Figure 5D*).

To examine whether the delayed phase transition in *cao/ch1* depends on miR156 function, we crossed *35S::MIM156* into *cao/ch1*. Similarly to *35S::MIM156*, *35S::MIM156 cao/ch1* produced the abaxial trichomes on the first leaf, and the leaves were elongated and serrated (*Figure 5A–C*; *Figure 5—figure supplement 1*). Compared to the wild type, the *cao/ch1* mutants exhibited higher glucose sensitivity. Treatment of *cao/ch1* seedlings with 50 mM glucose significantly reduced the level of miR156 (*Figure 5E*). Taken together, we conclude that sugar from the pre-existing leaves acts as a mobile signal to trigger the juvenile-to-adult phase transition through repression of miR156 in the young leaf primordia.

## Repression of miR156 by sugar is evolutionarily conserved

miR156 is present in all major plant taxa (*Axtell and Bowman, 2008*). To test whether the regulation of miR156 by sugar is evolutionarily conserved, we examined the expression of miR156 in response to

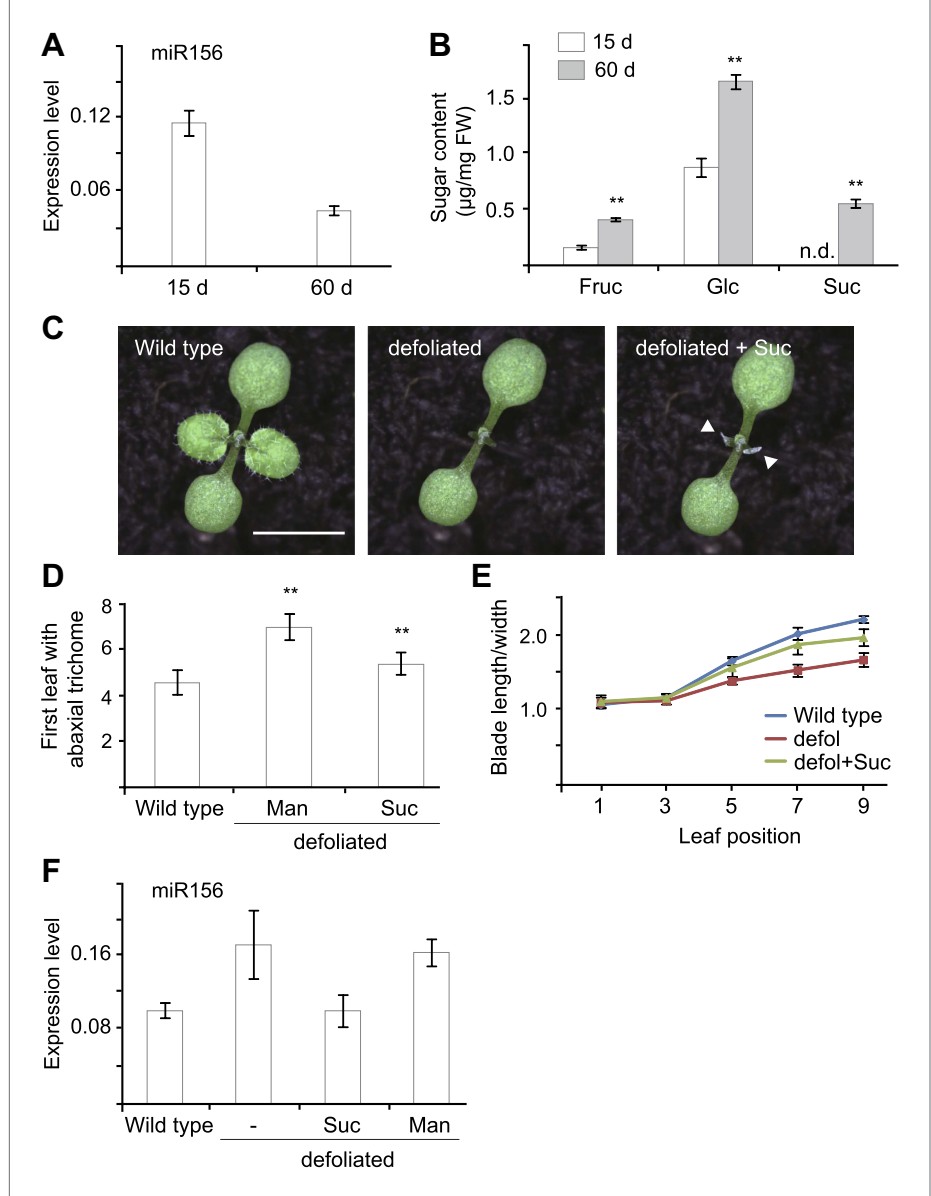

**Figure 4**. Sugar as a mobile signal to trigger vegetative phase transition. (**A**) Expression of miR156 in 15-day-old and 60-day-old wild type plants grown under short day conditions. (**B**) Sugar measurement. Fifteen-day-old and 60-day-old short day plants were collected at Zeitgeber time 16. The fructose (Fru), glucose (Glc), and sucrose (Suc) content was analyzed by GC-MS and quantified. **Significant difference from 15-day-old wild type plants, Student t-test, p<0.001. Error bars indicate SD. n.d.: undetected; FW: fresh weight. (**C**) Seven-day-old wild type Arabidopsis seedlings before and after defoliation. Arrows indicate where the lanolin-sucrose (Suc) paste was applied. Scale bar indicates 0.5 cm. (**D** and **E**) Seven-day-old wild type seedlings before and after defoliation. Appearance of the first abaxial trichome (**D**) and the length-to-width ratios of blades (**E**) were measured. n=10. **Significant difference from wild type, Student t-test, p<0.001. Error bars indicate SE. defol: defoliated; Suc: sucrose. (**F**) Expression of miR156. Seven-day-old wild type seedlings were defoliated and sucrose (Suc) or mannitol (Man) was applied to the defoliated petioles. The shoot apices were collected for expression analyses 2 days after defoliation.

sugar in other plants, including *Nicotiana benthamiana* (tobacco), *Physcomitrella patens* (moss), and *Solanum lycopersicum* (tomato).

*N. benthamiana* and *S. lycopersicum* were grown in 1/2 MS liquid media without sugar. After the first two leaves appeared, the seedlings were treated with 50 mM sucrose for 2 days. The seedlings of

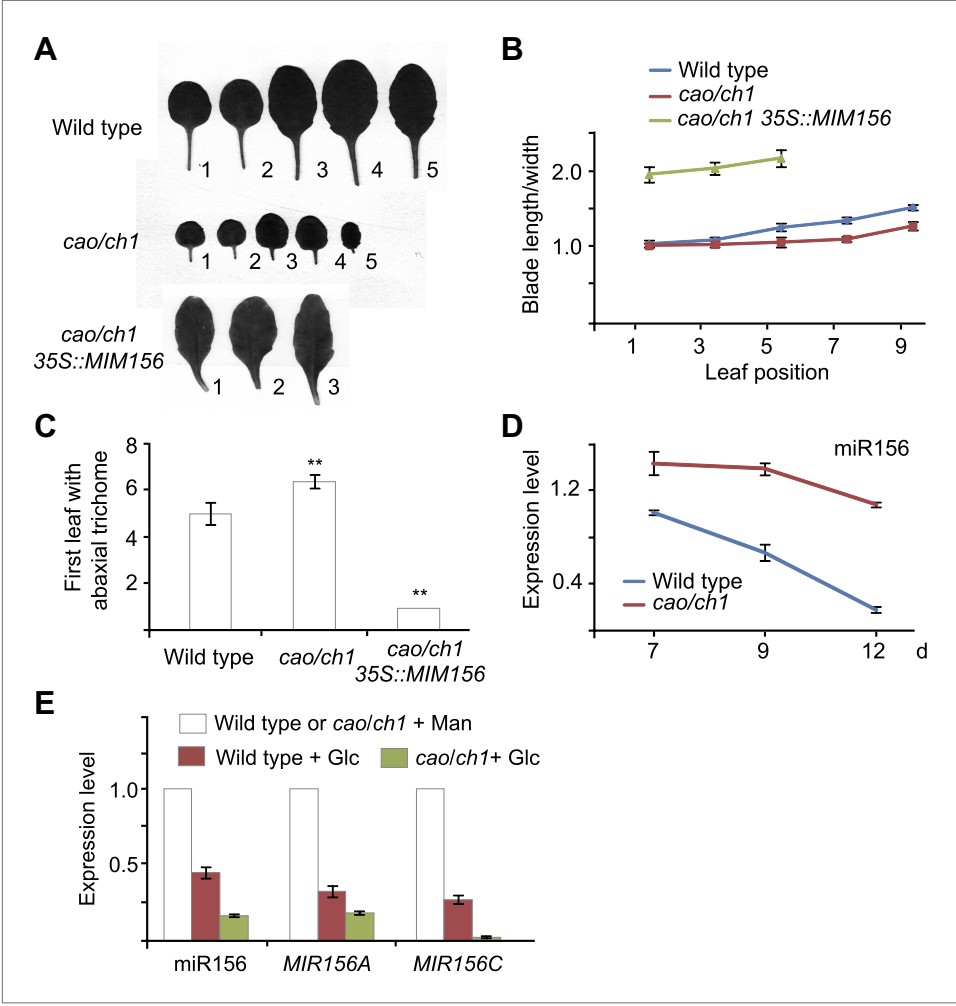

**Figure 5**. *cao/ch1* mutant impairs vegetative phase transition. (**A**) Leaf morphology of wild type, *cao/ch1*, and *35S::MIM156 cao/ch1* plants under long day conditions. The leaves from 15-day-old plants were detached and scanned. The numbers indicate leaf positions. (**B** and **C**) The length-to-width ratio of the blade (**B**) and the appearance of the first abaxial trichome (**C**). n=12. (**D**) Expression of miR156 during development. Wild type plants and *cao/ch1* mutants were collected at 7, 9, or 12 days after germination under long day conditions. (**E**) Expression of miR156, *pri-MIR156A*, and *pri-MIR156C*. Five-day-old wild type and *cao/ch1* mutants in 1/2 Murashige and Skoog (MS) liquid media were treated with 50 mM glucose or mannitol for 1 day. The expression levels in the mannitol-treated wild type or *cao/ch1* were set to 1. The treatment was started at Zeitgeber time 12. **Significant difference from wild type, Student t-test, p<0.001. Error bars indicate SE.

The following figure supplements are available for figure 5:

**Figure supplement 1**. Phenotype of *cao* mutant.

---

*N. benthamiana* and *S. lycopersicum* were collected and used for expression analyses. For *P. patens*, the sugar treatment was conducted during the protonema stage. Compared to those treated with mannitol, the amount of miR156 was greatly reduced in all the sucrose-treated plants (*Figure 6*), indicating that repression of miR156 by sugar is evolutionarily conserved.

## Sugar regulates miR156 expression at both the transcriptional and post-transcriptional level

To investigate at which level sugar represses miR156, we performed chromatin immunoprecipitation analyses (ChIP) using anti-RNA polymerase II (anti-Pol II) antibody, which recognizes the C-terminal heptapeptide repeat of RNA Pol II and has been used to correlate RNA Pol II binding with gene

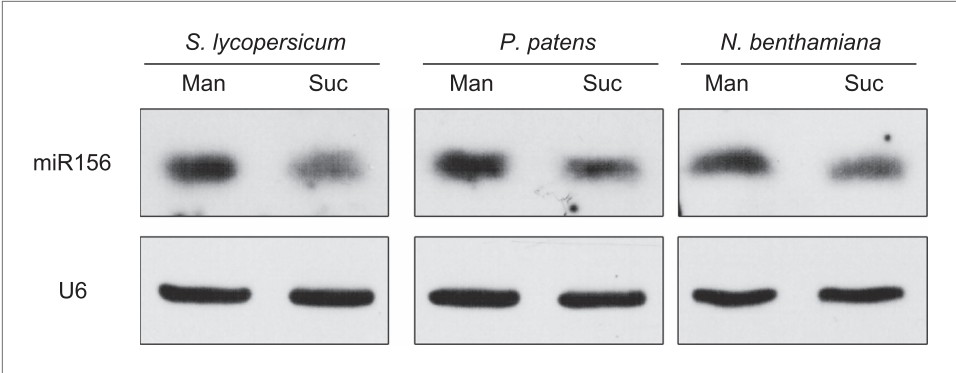

**Figure 6**. Repression of miR156 by sugar is evolutionarily conserved. Expression of miR156 in *Physcomitrella patens*, *Solanum lycopersicum*, and *Nicotiana benthamiana*. The plants were treated with 50 mM sucrose (Suc) or mannitol (Man) for 2 days. U6 was monitored as the loading control. Treatment was started at Zeitgeber time 12.

expression. Enrichment of the promoter fragments of *MIR156A* and *MIR156C* was compared between the seedlings treated with mannitol and those treated with glucose. As shown in *Figure 7A*, the promoter fragments (harboring TATA boxes) of *MIR156A* and *MIR156C* were substantially enriched in the mannitol-treated seedlings, but not in those treated with glucose, indicating that glucose induces transcriptional repression of *MIR156A* and *MIR156C* (*Figure 7A*).

We next examined the effect of actinomycin-D (ActD), which blocks transcription. To test transcription blocking efficiency, we analyzed the expression of *HXK1*, which is rapidly induced by glucose (*Price et al., 2004*). The transcript level of *HXK1* was increased about fourfold after 3 h of glucose treatment. By contrast, the expression of *HXK1* was not altered in the seedlings treated with glucose and ActD (*Figure 7B*).

The addition of ActD did not affect repression of *pri-MIR156C* by glucose. The transcript level of *pri-MIR156C* was reduced by about 75% after 3 hr in the presence of glucose, compared to a 30% reduction in the presence of mannitol (*Figure 7D*). A similar expression pattern was observed in *pri-MIR156A* (*Figure 7C*), suggesting that glucose modulates miR156 expression at the post-transcriptional level through the degradation of *pri-MIR156*.

To investigate whether the reduction in miR156 primary transcripts after glucose treatment was caused by an increase in the processing efficiency of *pri-MIR156*, we performed the glucose treatment assay using the *serrate* (*se*) mutant which is defective in miRNA biogenesis (*Grigg et al., 2005*; *Lobbes et al., 2006*; *Yang et al., 2006*; *Laubinger et al., 2008*). Similar to the wild type, the amount of *pri-MIR156A* and *pri-MIR156C* was markedly decreased in the ActD/glucose-treated *se*-3 mutant (*Figure 7C,D*), indicating that glucose regulates the abundance of *pri-MIR156* independently of the miRNA processing machinery.

*HXK1* encodes a glucose sensor that transduces diverse sugar signals. *gin2*-1, the HXK1-null mutant (*Moore et al. 2003*), exhibited a lower level of miR156 than the wild type (*Figure 8A*). The expression of miR156 still decreased over time in the *gin2*-1 mutant (*Figure 8B*). To test whether the repression of miR156 by sugar is mediated by HXK1, we compared the glucose response between the wild type and the *gin2*-1 mutant. The expression of *pri-MIR156A* and *pri-MIR156C* was reduced after sugar treatment in both the wild type and the *gin2*-1 mutant (*Figure 8C,D*). Similarly, an evident decrease in *pri-MIR156C* was observed in the *gin2*-1 seedlings treated with ActD/glucose (*Figure 8E*). These results suggest that HXK1 plays a role in miR156 expression but is not absolutely required for the repression of miR156 by sugar.

We then performed the sugar treatment assay in the presence of ActD and cycloheximide (CHX), an inhibitor of protein synthesis. The level of *pri-MIR156C* transcripts was greatly reduced in the ActD-treated samples, but not in those treated with both ActD and CHX (*Figure 7—figure supplement 1*), suggesting that sugar-induced *pri-MIR156C* degradation requires de novo protein synthesis.

mRNAs can be degraded through several partially independent pathways, including nonsense-mediated mRNA decay (NMD), 5'-to-3' mRNA degradation via exonucleases, and 3'-to-5' mRNA

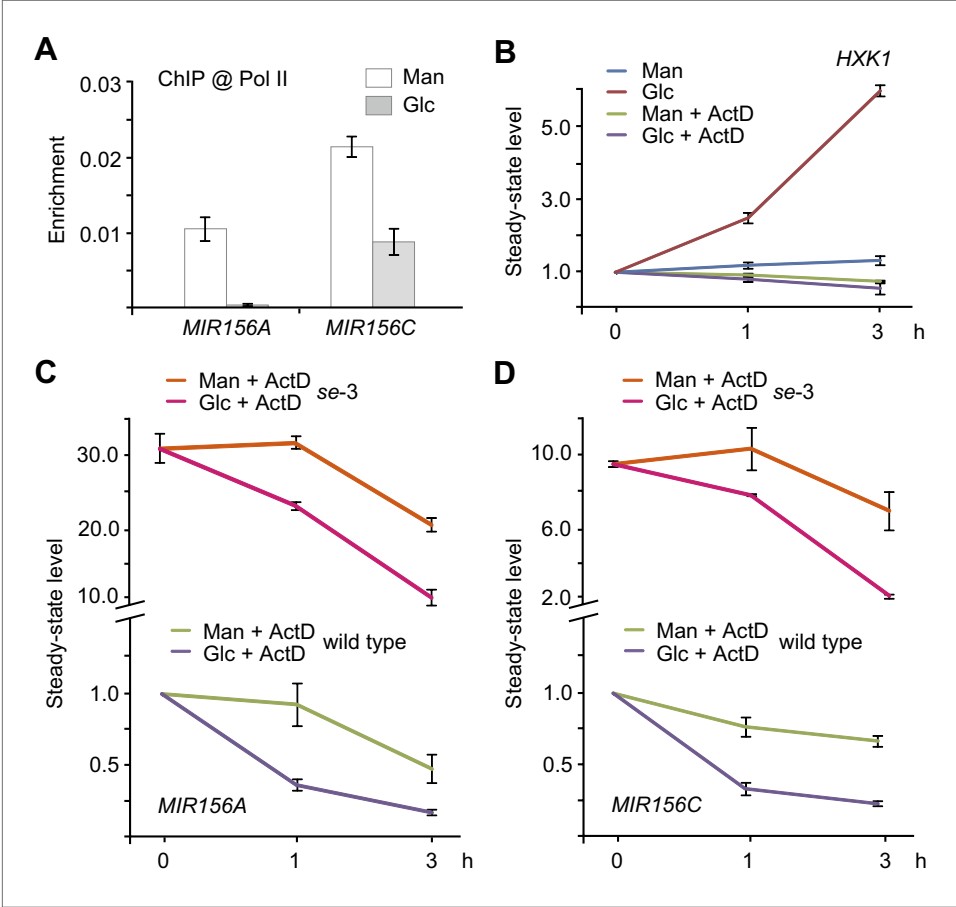

**Figure 7**. Sugar promotes the degradation of miR156 primary transcripts. (**A**) Chromatin immunoprecipitation (ChIP) analyses. Five-day-old wild type seedlings were treated with 50 mM glucose (Glc) or mannitol (Man) for 1 day. Anti-Pol II was used for ChIP analyses. The genomic fragments near the *MIR156A* or *MIR156C* TATA box were amplified. Relative enrichment was calculated by the ratio of bound DNAs after ChIP to input DNAs. (**B**) Expression of *HXK1* in response to glucose. Five-day-old wild type seedlings in 1/2 Murashige and Skoog (MS) liquid media were pre-treated with or without actinomycin (ActD) for 12 hr. The seedlings were harvested at 0, 1, and 3 hr after 50 mM glucose or mannitol was added. The expression level at 0 hr was set to 1. (**C** and **D**) Expression of *pri-MIR156A* (**C**) and *pri-MIR156C* (**D**) in the wild type and *se-3* mutant. Five-day-old wild type seedlings in 1/2 MS liquid media were pre-treated with ActD for 12 hr. The seedlings were then treated with 50 mM glucose or mannitol. The expression levels of *pri-MIR156A* and *pri-MIR156C* in the wild type at 0 hr were set to 1. Sugar treatment was started at Zeitgeber time 12.

The following figure supplements are available for figure 7:

**Figure supplement 1**. Effect of CHX on sugar-induced *pri-MIR156C* degradation.

**Figure supplement 2**. Expression analyses of *pri-MIR156A* and *pri-MIR156C* in *upf* mutants.

degradation via the exosome. The UP-frameshift (UPF) proteins, UPF1, UPF2, and UPF3, are essential for the NMD function in plants (**Arciga-Reyes et al., 2006**; **Kurihara et al., 2009**). It has been shown that *upf1* and *upf3* mutants impair the sugar response and over-accumulate sugar-inducible mRNAs (**Yoine et al. 2006**). Therefore, we investigated the role of UPF in the sugar-mediated repression of miR156. Compared to the wild type, the expression of *pri-MIR156A* and *pri-MIR156C* was slightly increased in *upf1*-5 and *upf3*-1 mutants (**Figure 7—figure supplement 2A**). Glucose was still able to repress the accumulation of *pri-MIR156C*, albeit to a lesser extent (**Figure 7— figure supplement 2B**), indicating that sugar promotes *pri-MIR156* degradation independently of canonical NMD.

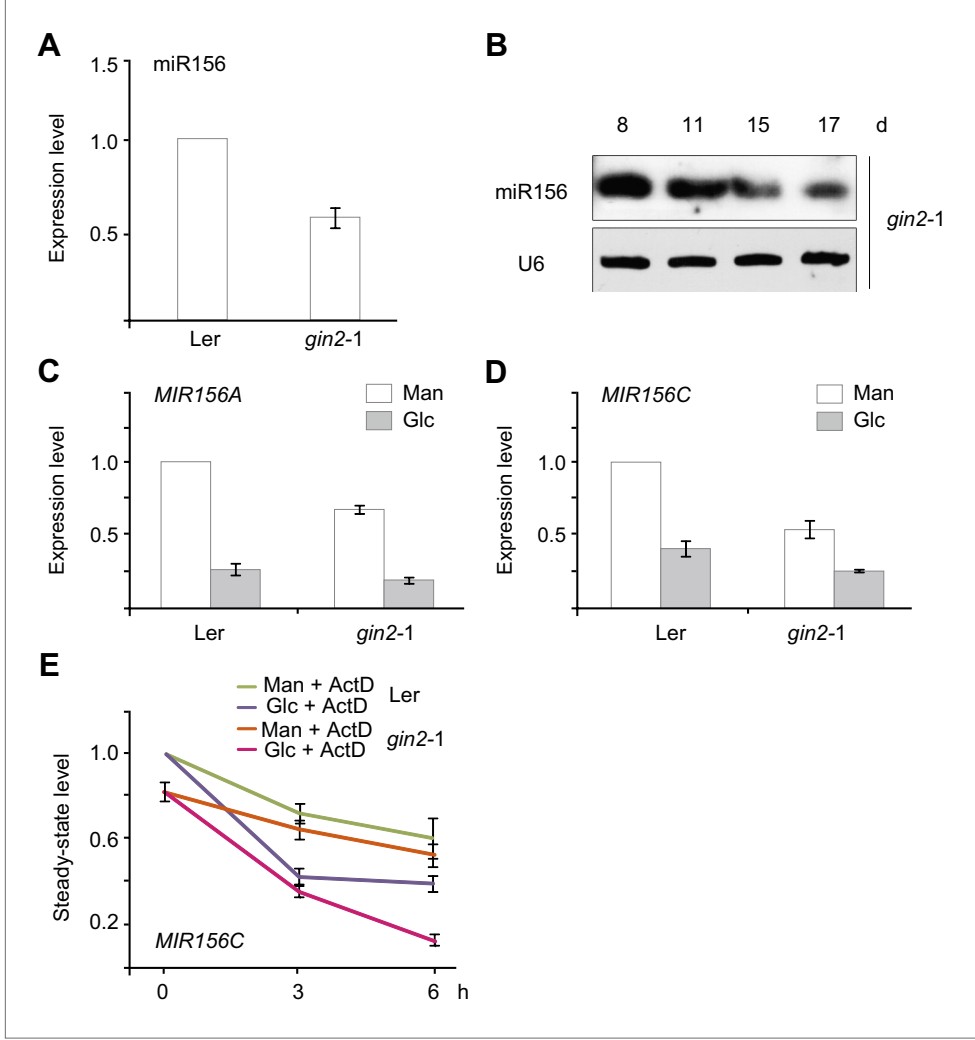

**Figure 8**. The role of *HXK1* in sugar-induced miR156 repression. (**A**) Expression of miR156 in the 5-day-old wild type (ecotype Ler) and *gin2*-1 mutant. The expression level of miR156 in Ler was set to 1. (**B**) Time course analyses of miR156 in the *gin2*-1 mutant. (**C** and **D**) Expression of *pri-MIR156A* (**C**) and *pri-MIR156C* (**D**) in response to glucose in the wild type (ecotype Ler) and *gin2*-1 mutant. Five-day-old seedlings in 1/2 Murashige and Skoog (MS) liquid media were treated with 50 mM glucose (Glc) or mannitol (Man) for 6 hr. The expression level in Ler at 0 h was set to 1. (**E**) Expression of *pri-MIR156C* in Ler and *gin2*-1. Five-day-old seedlings in 1/2 MS liquid media were pre-treated with actinomycin-D (ActD) for 12 hr and then treated with 50 mM glucose or mannitol. The expression level of *pri-MIR156C* in Ler at 0 hr was set to 1. Sugar treatment was started at Zeitgeber time 12.

## Discussion

### Sugar as an endogenous timer for the juvenile-to-adult phase transition

Based on expression analyses, defoliation experiments, and photosynthetic mutant characterization, we show that sugar acts upstream of miR156. We propose a model explaining how sugar regulates the juvenile-to-adult phase transition through modulation of miR156 expression as follows. After seed germination, plants start accumulating sugars through photosynthesis. Sucrose, the major transportable sugar, moves from the pre-existing leaves to the young leaf primordia, where its hydrolytic hexose product, glucose, represses the expression of miR156. As a result, the level of *SPL* increases and the expression of adult traits is promoted.

Identification of sugar as the endogenous developmental timing cue explains the irreversible nature of the age pathway. The level of miR156 is destined to decrease because the gradual accumulation of

carbohydrates is inevitable and essential for plant growth and development. In *Caenorhabditis elegans*, the transitions between the stages of larval development are controlled by the sequential action of two miRNAs, *lin-4* and *let-7* (*Pasquinelli and Ruvkun, 2002*; *Moss, 2007*; *Ambros, 2011*). In contrast to miR156, the expression of these two miRNAs is increased with age. It will be intriguing to examine whether sugar/carbohydrates or nutrients from the diet triggers the upregulation of *lin-4* and *let-7* in worms.

Cellular carbon (C) and nitrogen (N) are tightly coordinated to sustain optimal plant growth (*Raven et al., 2004*; *Zheng, 2009*). C compounds including many carbohydrates such as sucrose and glucose are synthesized in the leaf, while N nutrients such as nitrate ($NO_3^-$) and ammonium ($NH_4^+$) are assimilated by the root system. Biochemical and physiological studies have demonstrated long-distance sensing and signaling of the C/N balance in plants. When soil is short of $NH_4^+$ and $NO_3^-$, photosynthesis in the leaf is inhibited. Whether miR156 and the juvenile-to-adult transition respond to N excess or deficient conditions is another interesting topic awaiting investigation.

## Does sugar promote flowering through miR156?

In addition to the juvenile-to-adult transition, flowering is of great importance for reproductive success in plants. Previous studies revealed that the floral transition is regulated by diverse environmental factors, such as photoperiod, temperature, and light, in combination with the endogenous signal derived from nutritional status. The nutrient-dependent regulation of flowering is likely dependant on the rate of sucrose export from source leaves (*Corbesier et al., 1998*; *Sivitz et al., 2007*). This notion is supported by our observations that sugar from pre-existing leaves acts as a long-distance signal to repress the expression of miR156 in young leaf primordia, and that a high level of miR156 delays flowering (*Wang et al., 2009*). Intriguingly, a recent study has demonstrated that INDETERMINATE DOMAIN transcription factor AtIDD8 regulates photoperiodic flowering by modulating sugar transport and metabolism (*Seo et al., 2011*), suggesting that additional sugar-mediated flowering pathways exist.

Sugar, produced in mesophyll cells in leaves, is transported from source tissues to sink tissues through vascular bundles (*Kuhn and Grof, 2010*; *Ayre, 2011*). In Arabidopsis, sucrose transporters are involved in loading sucrose into the phloem in source leaves and the uptake of sucrose into the cells of sink tissues such as roots, fruit, and developing leaves (*Williams et al., 2000*). Very recently, the sucrose effluxers, SWEET11 and SWEET12, which facilitate sucrose efflux into the cell wall of companion cells, have been identified (*Chen et al., 2011*). It is therefore interesting to investigate whether impairment of sucrose transport from leaf cells into the vascular system causes a defect in miR156 expression and developmental transitions.

## Regulation of miR156 by sugar in a complex manner

There are several means by which sugar regulates gene expression. For example, sugar decreases the transcript level of rice *AMY3* at both the transcriptional and post-transcriptional level. It was shown that destabilization of the mRNAs of *AMY3* is mediated by its 3′ untranslated region (UTR) (*Chan and Yu, 1998*). Similarly, we found that *pri-MIR156A* and *pri-MIR156C* are subjected to transcriptional repression as well as transcript degradation in response to glucose. This two-level expression control by sugar might contribute to robust repression of miR156, which leads to irreversible transition from the juvenile to adult phase in plants. In Arabidopsis, HXK1 is a glucose sensor that transduces diverse aspects of sugar response. For example, the *gin2*-1 mutant reduces shoot and root growth, delays flowering, increases apical dominance, and alters sensitivity to auxin and cytokinin (*Moore et al., 2003*). However, we did not observe an obvious juvenile-to-adult phase phenotype in the *gin2*-1 mutant under long day conditions (data not shown). Further studies will determine if the transcriptional repression of miR156 by sugar is mediated by the previously identified nuclear-localized HXK1-VHA-B1-RPT5B complex.

The level of miR156 is greatly reduced when plants are treated with both glucose and sucrose. Since these sugars can be easily interconverted, it remains unclear whether the repression of miR156 is hexose or sucrose-dependent. Moreover, based on pharmacological treatment and mutant analyses, we show that sugar is able to trigger the degradation of *pri-MIR156A/C* independently of the canonical glucose sensor, HXK1. Thus, investigation of the molecular mechanism by which sugar in particular recognizes *pri-MIR156* and promotes their degradation is an important goal for future research.

## Materials and methods

### Plant materials

*A. thaliana*, *P. patens*, *S. lycopersicum*, and *N. benthamiana* were grown at 21°C (day)/19°C (night) under long day (16 hr light/8 hr dark) or short day (8 hr light/16 hr dark) conditions. White light was provided by a 4:2 mixture of cool white fluorescent lamps (Lifemax cool daylight 36W/865; Philips Lighting Co., Shangai, China) and warm white fluorescent lamps (Lifemax warm white 36W/830; Philips Lighting Co.). Light intensity was 80 µmol/m$^2$/s in long day and 90 µmol/m$^2$/s in short day conditions. *mir156a* (SALK_056809), *mir156c* (SALK_004679), *cao/ch1*, and *gin2*-1 mutants were ordered from the Arabidopsis Biological Resource Center (Columbus, OH). *35S::MIM156* was described (**Wang et al., 2008**).

### Plant treatment

All treatment assays were carried out under long day conditions. Defoliation assays were performed as described (**Yang et al., 2011**). For the sugar treatment assay, Arabidopsis seeds were sterilized with 20% bleach and germinated in 50 ml 1/2 MS liquid media with shaking at 140 rpm. The seedlings were then transferred to 1/2 MS media supplemented with sugar. For the sugar starvation assay, 5-day-old wild type seedlings grown in 1/2 MS liquid media supplemented with 50 mM sucrose were transferred to 1/2 MS liquid media free of sugar and grown in the dark for 2 days. For the ActD and CHX assay, 20 µg/ml ActD (Sigma-Aldrich, Beijing, China) or 100 µM CHX (Sigma-Aldrich) was used. *P. patens* was cultured as described (**Cove et al., 2009**). The plants in the protonema stage were used for the sugar treatment assay. Seedlings of *S. lycopersicum* and *N. benthamiana* were treated with 50 mM sucrose for 2 days.

### Expression analyses

Total RNA was extracted with Trizol reagent (Invitrogen, Life Technologies, Shanghai, China). Then 1 µg of total RNA was DNase I-treated and used for cDNA synthesis with an oligo (dT) primer. The qRT-PCR primers for *SPL3*, *SPL9*, *SPL15*, and *TUB* have been described (**Wang et al., 2008**; **Wang et al., 2009**). The primer sequences for other genes are shown in **supplementary file 1B**. A small RNA blot was performed as described (**Wang et al., 2009**). qRT-PCR on mature miR156 was performed according to a published protocol (**Varkonyi-Gasic et al., 2007**).

### ChIP analyses

ChIP analysis was performed according to protocol (**Wang et al., 2009**). Crude chromatin extract was pulled down with anti-Pol II antibodies (Abcam, Hong Kong, China). ChIP DNAs were reverse crosslinked and purified using a PCR purification kit (Qiagen, Shanghai, China). A 1 µl sample of DNA was used for real-time PCR analyses. The relative enrichment of each fragment was calculated by the ratio of bound DNAs after ChIP to input DNAs.

### Sugar measurement

Wild type plants were grown under short day conditions. Then 15-day-old juvenile or 50-day-old adult plants were collected at ZT 16. Sugar was measured using 50 mg (fresh weight) of tissue. Sample extraction, preparation, and analyses were performed as previously described (**Tan et al., 2011**). The individual sugar was identified based on the retention time and mass spectrometry standards. Quantification was performed by an external standard method.

## Acknowledgements

We thank Li Yang and Scott Poethig for the provision of unpublished data; Hong-Tao Liu for discussion; Han Xiao for *S. lycopersicum* seeds; and the Arabidopsis Biological Resource Center for *cao/ch1*, *gin2*-1, and *MIR156* T-DNA insertion mutants.

## Additional information

### Funding

| Funder | Grant reference number | Author |
|---|---|---|
| National Natural Science Foundation of China | 31222029; 91217306 | Jia-Wei Wang |

| Funder | Grant reference number | Author |
| --- | --- | --- |
| State Key Basic Research Program of China | 2013CB127000 | Jirong Huang, Jia-Wei Wang |
| Recruitment Program of Global Expects (China) | | Jia-Wei Wang |
| Shanghai Pujiang Program | 12PJ1409900 | Jia-Wei Wang |
| Initiation grant from National Key Laboratory of Plant Molecular Genetics (Institute of Plant Physiology and Ecology, Shanghai Institutes for Biological Sciences) | | Jia-Wei Wang |
| Chinese Academy of Sciences | KSCX2-YW-N-069 | Guodong Wang |

The funders had no role in study design, data collection and interpretation, or the decision to submit the work for publication.

## Author contributions

SY, HL, Acquisition of data, Analysis and interpretation of data; LC, Acquisition of data, Analysis of data (sugar measurement); T-QZ, Acquisition of data, Analysis of data (tobacco small RNA blot); C-MZ, Acquisition of data, Analysis and interpretation of data, Drafting or revising the article; YS, JW, JH, Analysis and interpretation of data, Contributed unpublished essential data or reagents; GW, Experiment design, Analysis and interpretation of data (sugar measurement); J-WW, Conception and design, Analysis and interpretation of data, Drafting or revising the article

# Additional files

## Supplementary files

• Supplementary file 1. (A) *MIR156* T-DNA insertion mutants. (B) Oligonucleotide primer sequences.

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
