## [Decision Letter]

Thank you for choosing to send your work entitled “Sugar is an Endogenous Cue for Developmental Timing in Plants” for consideration at eLife. Your article has been evaluated by a Senior editor (Detlef Weigel) and 2 reviewers, one of whom (Rick Amasino) is a member of our Board of Reviewing Editors.

The Reviewing editor and the other reviewer discussed their comments before we reached this decision, and the Reviewing editor has assembled the following comments based on the reviewers' reports.

The finding that sugar may be the age signal that acts via the miR156/SPL module is a significant advance in plant biology that will stimulate further research.

There are, however, some issues that need to be resolved, in particular the inconsistencies between your paper and the related paper submitted by the Poethig lab.

[Editors' note: these two studies were conducted independently; each was reviewed on its own merits, on the understanding that it would be published alongside the related study if both were accepted.]

Although the major conclusion of both papers is the same, namely that sugar treatment reduces the levels of miR156, three major differences are: 1) effect or lack thereof of cycloheximide, 2) hexokinase dependence (i.e., whether or not HXK1 is a glucose sensor for the age effect), and 3) transcriptional versus post-transcriptional regulation of *MIR156* expression. Some of these inconsistencies might result from studying different promoters (*MIR156A* vs *MIR156C*), measuring miR156 versus *pri-MIR156C* levels, the use of liquid culture versus solid media, or different temperature and light regimens.

Two other issues are:

4) The possible difference in effect of the *cao/chlorina1* mutant is due to short day versus long day conditions. In addition to day length, light intensity, and temperature are likely to play a role. (There also needs to be agreement on the nomenclature of the mutant.)

5) Differences in glucose concentrations used.

We hope that you will exchange information with Scott Poethig and colleagues on experimental design, repeat a number of crucial experiments under common conditions (which should not take long), and determine if the inconsistencies may in fact be resolvable. We will relay a corresponding message to the Poethig lab.

A final note: the editors and reviewers discussed the lack of measurement of endogenous sugars during phase change and concluded that, although this would be a valuable addition to the body of work you have presented, it would be beyond the scope of your study.

Comments specific to your paper:

A) The Materials and methods often do not provide sufficient detail for the reader to evaluate your work. A major example is the lack of detail of plant growth conditions. You simply note LD and SD. But for the type of research you are presenting – an effect of sugar – a detailed description of light intensity and light quality, as well as temperature is critical.

B) You should attempt to quantify GUS activity for the MIR156 promoter response experiments.

C) The ActD treatment should have a control (e.g., from another miRNA gene, which presumably should have no effect).

D) Another useful control would be measurements from wild-type plants alongside the sugar-response result for the *gin2-1* mutant. *gin2-1* still responds to sugar, but does it respond as much as wild type in the same experiment?

E) Removal of the first two leaves in 7d old Arabidopsis seedlings resulted in a delayed expression of adult traits and this can be partially overcome by sucrose application to the remaining petioles. This indicates that sugars may be the mobile signal. However, the effects of defoliation and subsequent sugar feeding on miR156 expression were not investigated, leaving part of the question unresolved.

F) In the *cao* mutant and in norflurazon-treated plants, it is important to determine whether all aspects of the vegetative phase change phenotype (morphological as well as molecular) can be rescued by application of glucose or sucrose. This is crucial, especially in the norflurazon experiment, as, contrary to what is claimed, not only are pre-existing leaves affected by the treatment, but the newly formed leaves are too, as can be seen in Figure 3H. Therefore it cannot be ruled out that there is a direct norflurazon effect on miR156 expression.

G) The title “Sugar is an endogenous cue for developmental timing in plants” is too general as only one aspect of developmental timing is studied in detail (vegetative phase change).

[Editors' note: the authors were also asked to address the following comment before acceptance.]

Before acceptance, it is necessary for you to compare Wt and *gin2* mutant directly because only the immediate response in the presence of ActD has been examined. Specifically you need to show the explicit comparison by presenting the data in Figure 7C+D, 7E+F, 7 G+H and normalized to the same standard. If another round of experiments are needed, these can be performed rapidly as you are working with seedlings.

---

## [Author Response]

*Although the major conclusion of both papers is the same, namely that sugar treatment reduces the levels of miR156, three major differences are: 1) effect or lack thereof of cycloheximide, 2) hexokinase dependence (i.e., whether or not HXK1 is a glucose sensor for the age effect), and 3) transcriptional versus post-transcriptional regulation of* MIR156 *expression. Some of these inconsistencies might result from studying different promoters (*MIR156A *vs* MIR156C*), measuring miR156 versus* pri-MIR156C *levels, the use of liquid culture versus solid media, or different temperature and light regimens*.

We have discussed our results with Scott Poethig. In the revision, most of the inconsistencies between our papers have been resolved. The conclusion is that sugar represses miR156 at both transcriptional level (partially dependent on HXK1), and post-transcriptional level through promoting degradation of *pri-MIR156* transcripts.

In the last version, we did not exclude the possibility that sugar regulates miR156 expression through transcriptional level. In the revision, we performed ChIP assay using anti-Pol II antibody. The enrichment of the genomic fragment of *MIR156A* and *MIR156C* was markedly reduced after glucose treatment, demonstrating that sugar is able to repress miR156 expression at transcriptional level (Figure 7A).

In the revised version, our paper focuses on how sugar represses miR156 at the post-transcriptional level in an HXK1-independent manner, while the paper from the Poethig lab emphasizes the role of sugar in the transcriptional control of *MIR156A*.

After blocking transcription by ActD, we found that sugar is able to repress miR156 through promoting the degradation of primary transcripts of *MIR156A* and *MIR156C*. This sugar-induced pri-miRNA degradation is not dependent on HXK1 but requires de novo protein synthesis.

*4) The possible difference in effect of the* cao/chlorina1 *mutant is due to short day versus long day conditions. In addition to day length, light intensity**, and temperature are likely to play a role. (There also needs to be agreement on the nomenclature of the mutant.)*

We did observe a defect in juvenile-to-adult phase transition in the *cao/ch1* mutant under our growth conditions. This inconsistency could be due to different growth conditions. In the revision, the nomenclature of the *cao* mutant has been unified by using *cao/ch1*.

*5) Differences in glucose concentrations used*.

The inconsistencies between our papers were not due to the different glucose concentration that we used. We repeated our experiment using 10 mM glucose and got the same results as those from Poethig lab.

*A final note: the editors and reviewers discussed the lack of measurement of endogenous sugars during phase change and concluded that, although this would be a valuable addition to the body of work you have presented, it would be beyond the scope of your study*.

According to your suggestions, we performed sugar measurements. We compared the sugar content between juvenile and adult plants. We could not obtain reliable results in long day, probably due to a rapid life cycle under this condition. We then performed our measurement in short day. It appears that the adult plants accumulated more sugar (glucose, fructose, and sucrose) than juvenile plants. Consistently, we found that there was a nice correlation between miR156 level and sugar content in these samples (Figure 4A and 4B).

Comments specific to your paper:

*A) The Materials and methods often do not provide sufficient detail for the reader to evaluate your work. A major example is the lack of detail of plant growth conditions. You simply note LD and SD. But for the type of research you are presenting – an effect of sugar – a detailed description of light intensity and light quality, as well as temperature is critical*.

We have included further details in the Materials and methods. For example: “*A. thaliana*, *P. patens*, *S. lycopersicum*, and *N. benthamiana* plants were grown at 21 °C (day)/19 °C (night) in long days (16 hours light/8 hours dark). White light was provided by a 4:2 mixture of cool white fluorescent lamps (Lifemax cool daylight 36W/865; Philips Lighting Co., China) and warm white fluorescent lamps (Lifemax warm white 36W/830, Philips Lighting Co.). Light intensity is 80 µmol/m^2^/s.”

*B) You should attempt to quantify GUS activity for the MIR156 promoter response experiments*.

We communicated with Scott Poethig. It turned out that the *MIR156A/C* promoter GUS reporters from Peter Huijser's lab lack the essential sugar responsive element in the 3'UTR. Therefore, we removed this result in the revision.

*C) The ActD treatment should have a control (e.g., from another miRNA gene, which presumably should have no effect)*.

We used HXK1 as a control (Figure 7B). HXK1 is rapidly induced by glucose. This effect is abolished after the addition of ActD, indicating that our treatment is effective.

*D) Another useful control would be measurements from wild-type plants alongside the sugar-response result for the* gin2-1 *mutant*. gin2-1 *still responds to sugar, but does it respond as much as wild type in the same experiment?*

*gin2-1* is in the Ler genetic background. Therefore, we performed the same treatment in wild-type Ler accession. There is no difference in miR156 expression between Ler and *gin2-1* (Figure 7E).

*E) Removal of the first two leaves in 7d old Arabidopsis seedlings resulted in a delayed expression of adult traits and this can be partially overcome by sucrose application to the remaining petioles. This indicates that sugars may be the mobile signal. However, the effects of defoliation and subsequent sugar feeding on miR156 expression were not investigated, leaving part of the question unresolved*.

We performed the expression analyses of miR156. Compared to intact plants, miR156 is increased after defoliation. This effect is suppressed by sucrose application (Figure 4F). This expression pattern is consistent with our phenotypic characterizations.

*F) In the* cao *mutant and in norflurazon-treated plants, it is important to determine whether all aspects of the vegetative phase change phenotype (morphological as well as molecular) can be rescued by application of glucose or sucrose. This is crucial, especially in the norflurazon experiment, as, contrary to what is claimed, not only are pre-existing leaves affected by the treatment, but the newly formed leaves are too, as can be seen in Figure 3H. Therefore it cannot be ruled out that there is a direct norflurazon effect on miR156 expression*.

We agree with this argument. Therefore we removed the norflurazon treatment assay in the revision.

*G) The title “Sugar is an endogenous cue for developmental timing in plants” is too general as only one aspect of developmental timing is studied in detail (vegetative phase change)*.

We have changed it to “Sugar is an endogenous cue for juvenile-to-adult phase transition in plants”.

*Before acceptance, it is necessary for you to compare Wt and* gin2 *mutant directly because only the immediate response in the presence of ActD has been examined. Specifically you need to show the explicit comparison by presenting the data in Figure 7C+D, 7E+F, 7 G+H and normalized to the same standard. If another round of experiments are needed, these can be performed rapidly as you are working with seedlings*.

We compared the expression of miR156 and *pri-MIR156* between Wt and *gin2-1* (Figure 8). Compared to Wt, the *gin2-1* mutant has a lower level of miR156 in seedlings. However, the *gin2-1* mutant still exhibited the gradual decreased expression pattern of miR156 as Wt. Consistently, the expression of *pri-MIR156* was reduced in response to sugar treatment. We normalized the data to the same standard (Figure 7C, 7D, and 8E).